# Towards Sustainable Shifts to Healthy Diets and Food Security in Sub-Saharan Africa with Climate-Resilient Crops in Bread-Type Products: A Food System Analysis

**DOI:** 10.3390/foods11020135

**Published:** 2022-01-06

**Authors:** Martijn W. J. Noort, Stefano Renzetti, Vincent Linderhof, Gerrie E. du Rand, Nadéne J. M. M. Marx-Pienaar, Henriëtte L. de Kock, Nomzamo Magano, John R. N. Taylor

**Affiliations:** 1Wageningen Food & Biobased Research, Wageningen University & Research, P.O. Box 17, 6700 AA Wageningen, The Netherlands; stefano.renzetti@wur.nl; 2Wageningen Economic Research, Wageningen University & Research, P.O. Box 29703, 2502 LS The Hague, The Netherlands; vincent.linderhof@wur.nl; 3Department of Consumer and Food Sciences, University of Pretoria, Private Bag X20 Hatfield, Pretoria 0028, South Africa; gerrie.durand@up.ac.za (G.E.d.R.); nadene.marxp@up.ac.za (N.J.M.M.M.-P.); riette.dekock@up.ac.za (H.L.d.K.); zamonthabi@gmail.com (N.M.); john.taylor@up.ac.za (J.R.N.T.)

**Keywords:** food system, cereals, pulses, wheat, bread, climate, food and nutrition security

## Abstract

Massive urbanization and increasing disposable incomes favor a rapid transition in diets and lifestyle in sub-Saharan Africa (SSA). As a result, the SSA population is becoming increasingly vulnerable to the double burden of malnutrition and obesity. This, combined with the increasing pressure to produce sufficient food and provide employment for this growing population together with the threat of climate change-induced declining crop yields, requires urgent sustainable solutions. Can an increase in the cultivation of climate-resilient crops (CRCs) and their utilization to produce attractive, convenient and nutritious bread products contribute to climate change adaptation and healthy and sustainable diets? A food system analysis of the bread food value chain in SSA indicates that replacement of refined, mostly imported, wheat in attractive bread products could (1) improve food and nutrition security, (2) bring about a shift to more nutritionally balanced diets, (3) increase economic inclusiveness and equitable benefits, and (4) improve sustainability and resilience of the food system. The food system analysis also provided systematic insight into the challenges and hurdles that need to be overcome to increase the availability, affordability and uptake of CRCs. Proposed interventions include improving the agronomic yield of CRCs, food product technology, raising consumer awareness and directing policies. Overall, integrated programs involving all stakeholders in the food system are needed.

## 1. Introduction

In sub-Saharan Africa (SSA), rural communities traditionally prepare meals from locally grown crops like cassava, sorghum and pulses. However, with rapid population growth, massive urbanization and increasing disposable incomes, consumption of refined wheat breads is rapidly increasing and displacing traditional meals. Major economic and food and nutrition security problems are resulting from this transition. The nutritional double burden is prevalent throughout SSA, with urban and peri-urban regions being most vulnerable, and is certain to increase as the nutrition transition advances [1,2]. Africa now imports nearly 60% of its wheat, with, for example, Kenya and Uganda importing some 68% and 95%, respectively, of their domestic needs. There are opportunities for Africa to reduce its dependency on wheat imports by replacing wheat flour with flours made of locally grown crops. This change is needed to improve food security, provide markets and regular income to smallholder farmers and create new business opportunities along the crop value chain. Furthermore, the transition of SSA agriculture to producing crops that are more resilient to abiotic stresses like drought, heat and flooding are crucial in the face of the climate change predicted for shorter- and longer-term future. A summary of the agronomic characteristics of some major climate-resilient crops (CRCs) compared to the major cereal crops is shown in Appendix A.

CRCs comprise food crops classified as cereals (e.g., sorghum, fonio, teff and finger millet), pseudocereals (amaranth), roots and tubers (cassava and sweet potato), pulses (*Phaseolus* beans, cowpeas, chickpeas, pigeon peas and Bambara groundnuts) and oilseed legumes (soya beans and peanuts). All these crops are widely grown across SSA, obviously depending on the local conditions. Many CRCs such as sorghum, finger millet, pearl millet, fonio, teff, Bambara groundnuts and cowpeas are indigenous food crops in SSA. Other major food crops like maize, cassava and *Phaseolus* beans originated in the Americas and were later introduced to Africa. The SSA communities that cultivate these crops have developed a very wide range of food products from them, ranging from dry grain snacks to porridges and gruels and to steamed and fried doughs and flatbreads (both fermented and unleavened). Wheat bread-type products and widespread use of wheat in SSA only started during the colonial times. Since then, wheat, which is mostly imported, has become one of the most important staples in SSA. The factors responsible for the trend towards increasingly using wheat as a staple are complex and multifaceted and therefore difficult to reverse. In this context, the study of food systems has taught us that change cannot be accomplished by individual policies, innovations or measures alone. To make a substantial change, a fundamental transformation in the food system is needed. It requires a thorough analysis of the entire food system (see Figure 1), which is the purpose of this paper.

The food system approach was formulated as a method to oversee the complexity of the entire food system, not merely limited to the food supply chain and its actors, but also taking into account the socioeconomic and environmental drivers which influence the food supply chain [3]. Furthermore, it also serves as a tool to oversee the outcomes of the food system with respect to its impacts on food security, health, socioeconomic and environmental aspects. Hence, this methodology is widely accepted and adopted as a logical framework to analyze the entire food system.

Figure 1 provides an overview of the food system framework. The food supply system comprises all the activities associated with food production and food utilization; it includes the agricultural production, storage, transport and trade, food processing, marketing and retail and finally food consumption and disposal of food waste. These food supply activities rely on socioeconomic drivers like labor and capital while providing socioeconomic outcomes like income and employment next to food security. However, they are also influenced by market developments, governmental policies and legislation. Finally, environmental factors also drive—and are affected by—the food supply system; directly, by the required input of soil, minerals, water and energy, but also by the climate itself and food supply impacts on the environment and biodiversity. By overseeing the food system entirely, the overall outcomes towards food security, safe and healthy diets, inclusiveness and equal benefits and sustainability and resilience can be evaluated [3].

In this study, we applied a food system approach focusing on the bread food value chain in SSA to identify the key success factors as well as challenges that need to be addressed to bring about the widespread use of CRCs in breadmaking in the region with the aim of improving food and nutrition security.

## 2. Analysis of the Food System

### 2.1. Socioeconomic Drivers

#### 2.1.1. SSA Agrifood Economy

Agricultural produce is one of the most important industries and trade of SSA [4], with African consumers spending on average over half of their expenditure on food. Traditionally, agricultural exports have been of major economic importance. However, in the last decades, agricultural productivity in SSA could not keep pace with the increasing internal demands due to population growth [4,5]. This resulted in SSA changing from a net agricultural exporter to a net agricultural importer [6,7]. The agricultural productivity growth in SSA has been slow since 2000 [8] and remained significantly below what could be achieved by using the best available practices and technology. According to a study of the Food and Agriculture Organization of the United Nations, the limited gains in productivity and competitiveness can be attributed to poor infrastructure, lack of productive technologies, lack of access to inputs and weak institutions [6]. Figure 2 shows the production volumes for some of the main starchy crops in SSA in comparison with SSA population growth. Since the increase in production of these crops cannot keep up with population growth, it drives the increasing importation of wheat and rice demonstrated by the clear parallel trends. 

The increasing demand for food offers opportunities for the agri-food value chain in SSA, but large-scale farmers are best-placed to take advantage of the new opportunities, while smallholder farmers are at risk of exclusion if they are not able to meet the demands of high-value markets [9]. Primary agricultural production offers only limited added value, and smallholders, in particular, benefit disproportionately little from the total economic benefit in the agri-food chain. The major part of added value lies in the upstream processing of food products and supply to the consumer. In high-income countries, the shift from agricultural production to food processing and distribution has been made, and big consumer food brands and retail chains control the chain and profit most. This transition to modern value chains offers an important opportunity for the SSA’s economy to grow. However, at the same time, smallholders and small and medium enterprises (SMEs) will need to be able to comply with increasing food safety standards to benefit from these opportunities.

**Figure 2 foods-11-00135-f002:**
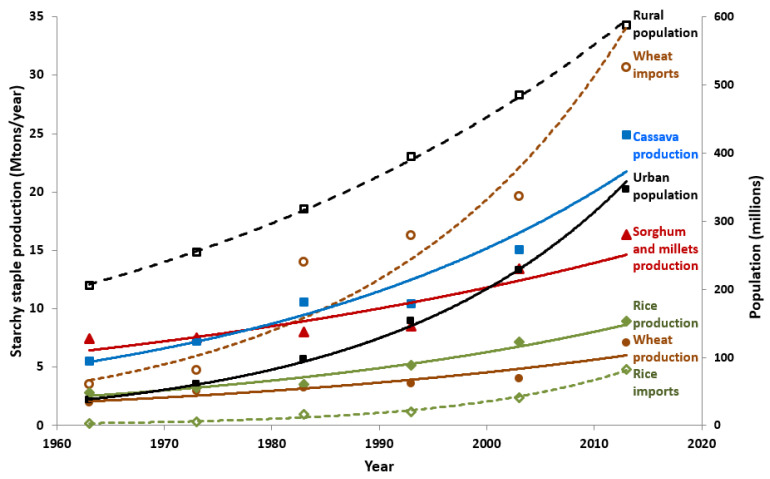
Trends in the production and imports of starchy staples and population size in sub-Sharan Africa from 1963 to 2013 [10].

#### 2.1.2. SSA Consumer Market Developments

The SSA market can be characterized as informal, complex and highly fragmented, in terms of the strongly diversified consumer segments and food supply channels as well as the huge local variations [5]. The SSA urban food market is in a transition phase towards increasing numbers of modern distribution outlets, such as supermarkets, self-service grocery stores and formal stores [11,12]. However, informal markets and street foods still constitute the largest part of daily urban food consumption in SSA. In view of the importance of street food, food technology innovations as well as product development and market introduction should be compatible with the capacities and demands of small local food processors and street vendors. In rural areas, the provision of food still mostly relies on home preparation of own produce or local food processors.

About 42% of Africans still live below the US $1.25 income per day poverty line, and one in four people in SSA remain undernourished [13]. Moreover, Africa has the highest gap between average incomes of the top 10% and the bottom 50%, with those in the 10% earning thirty times more than those in the bottom 50% [14]. A positive development, however, is that as a result of rising living standards over the past decade, the lower-middle class group is now the fastest expanding segment. This group, which is the most rapidly urbanizing, can be seen as the critically important consumer segment for market developments and consumption-driven economic growth in SSA [15]. A challenge is that this middle class is a highly heterogeneous consumer segment. Their cultural, ethnic, societal differences and divergent living conditions make it impossible to describe the group well in terms of shared consumer demands. Therefore, the SSA middle class cannot be compared to that of other parts of the world. In view of its socioeconomic and geographical fragmentation, reaching this consumer market requires a customized and city-centric consumer approach with a particular focus on intermediate cities for (food as well as other) product innovations [15]. 

SSA has a predominantly young population, with 65 percent of the population below the age of 35 [16]. This population segment is extremely important to consider with regard to (food) consumption, nutritional transition, (future) employment and their interactions [17]. Due to the young population of SSA, over 300 million new young employment seekers will arise in the next decade. Together with the ongoing urbanization and growing disposable incomes, a large shift from primary agri-business to the food processing chain is expected, both in terms of demand for employment as well as in the growing demand for processed foods to consume.

#### 2.1.3. Food Demand, Choice and Changing Dietary Patterns

Over the past 50–60 years, there have been profound changes in the types of staple food products consumed in SSA. Consumption of refined wheat breads is increasing rapidly and displacing traditional meals based on local crops, for example, porridges made from sorghum, millets and tubers [18]. Related consumer demands and behavior are further discussed in Section 2.3.4.

SSA has the world’s highest total population growth and urban population growth rates. Urbanization and its associated lifestyle changes, such as formally employed women, greatly increase the demand for quick-to-prepare convenience-type staple foods to feed their families [18]. These convenience-type staples are mostly based on wheat, such as leavened bread, flatbreads, noodles and pasta, and also on maize and rice [19,20]. This dietary transition from traditional foods to wheat-based bread products is attributed to a significant change in consumer attitudes which results from circumstantial factors such as changing lifestyles, demographics, urbanization, increased product availability, accessibility and technological changes [21]. Urbanization and modernization in SSA result in the adoption of Western lifestyles and trends that involve diverting from traditional food practices, food ingredients and, ultimately, eating patterns [22]. This dietary shift often leads to a less healthy diet and increased risk of nutrient deficiency since refined wheat products are rich in starch, fat and sugars but low in proteins, micronutrients and particularly in dietary fiber (see Section 2.4.2).

#### 2.1.4. Agricultural Sciences and Technology

Globally, the development of crops (breeding) as well as agronomical practices (fertilizers, pesticides, etc.) has resulted in an enormous increase in agricultural productivity. However, these developments have focused primarily on improving productivity of the major staple crops, such as wheat, rice, maize, soya beans and to a lesser extent the CRCs cassava and sorghum. This unbalanced development has contributed to a shift in production from indigenous and traditional crops to these global commodity crops. In the case of SSA, this resulted mainly in a very large increase in production of maize and cassava [23]. The gap in productivity and availability of cheap commodities versus local and indigenous crops has marginalized their cultivation to food security or subsistence crops or cultivation merely in restricted agronomic conditions (e.g., arid conditions, poor soils). For example, millets, which were traditionally the major food staples in rural SSA, have only shown a moderate increase in production in SSA despite the rapid population growth (Figure 2), which clearly indicates that these traditional crops are declining in their contribution to the diet. This underdevelopment is not limited only to primary production. Throughout the food supply chain, the global crops have catalyzed the attention towards scientific and technological developments, scale and uniformity in terms of processes and product applications, thus further limiting the valorization of CRCs.

#### 2.1.5. Standardization

Compliance to food standards is essential to ensure quality and safety throughout the food supply chain. As with wheat, there are specific Codex Alimentarius and national standards for CRCs and also for their respective flours. However, in contrast to large commodities, many CRCs, with the exception of some sorghum and cassava production, are generally produced at small scale by smallholder farmers. It is therefore problematic to ensure CRCs consistently meet these basic standards. For the use of CRCs in high-value applications like export, organic or gluten-free foods, additional standardization and certification are required, which is often not feasible for smallholder farmers. Furthermore, for further processing CRCs, good manufacturing practices (GMP) and hazard analysis and critical control points (HACCP) management systems are required to ensure food quality and safety. For small enterprises to initiate CRC-based food manufacturing, this is a major hurdle to overcome, and even more so if foods need to comply with the private standards of the major food retail and processing companies in Africa. These hurdles increase further if CRC-based crops, ingredients or foods are to be exported as they need to meet the legislative requirements of the EU and other developed countries and especially the private standards of the major internal food retail and processing companies.

#### 2.1.6. Governmental Policies and Legislation

National governments of the SSA countries and the FAO have recognized the disadvantages of the dietary shift towards refined wheat-based breads, and several intervention programs have been undertaken in the last 50–60 years [24,25,26,27,28,29]. These interventions have consisted of voluntary or mandatory inclusion of local crops with wheat flour (so-called composite flours) or even a total ban on wheat imports. Unfortunately, these wheat flour replacement initiatives have had limited success and impact for many reasons. An overview of these reasons is provided in Table 1, which clearly points out the multifaceted problems, which all need to be faced effectively. The experiences of these initiatives showed that constantly changing policies of individual governments are not able to provide structural changes throughout the food supply chain. It is clear that structural and regional (African) policies are needed to steer production and pricing of agricultural commodities to stimulate CRC production and improve their economic position compared to imported wheat. At the same time, the provision of financial incentives (i.e., tax rebates for CRCs and/or tax duties on imported wheat) itself is insufficient to steer to the use of CRCs throughout the food supply chain. This underlines the need for an innovative, more holistic approach and interventions based on a thorough understanding of the entire food system as we attempt to provide in this paper.

### 2.2. Environmental Drivers

#### 2.2.1. Climate

Climate change affecting weather patterns, notably higher temperatures and more frequent and severe droughts in the semiarid tropical regions like Southern and East Africa [30], is predicted to have a major impact on agricultural production systems and crop productivity. The Intergovernmental Panel on Climate Change (IPCC) reported that the SSA food system is among the world’s most vulnerable for climate change because of its’ extensive reliance on rainfed crop production, high intra- and interseasonal climate variability and recurrent droughts and floods [31]. In addition, persistent poverty is noted to limit the capacity of the region to adapt to these challenges.

Climate change is predicted to have an overall negative effect on the yields of the major cereal crops across Africa, with strong regional variability. For example, overall maize productivity in SSA is predicted to decline by 18–30% by 2050. Local exceptions are expected, e.g., in Eastern Africa where maize production could benefit from warming at high-elevation locations [31]. 

Sorghum is the most drought- and high temperature-tolerant cereal species and can have a yield and economic advantage over maize in dryland agriculture where water stress is a major problem [32]. Pearl millet has better high temperature tolerance than many other cereals [33], whereas amaranth is particularly drought-tolerant [34]. Due to its high climate change resilience, sorghum is predicted to lose only 2% in yield, compared to wheat expected to lose as much as 35% in productivity by 2050. In West Africa, temperature and precipitation increase are even estimated to have positive effects on millet and sorghum yields [31]. 

Cassava is considered to be climate-resilient as it can have better tolerance to high temperatures and variable precipitation in certain environments than cereals [35]. In general, the IPCC projects cassava yields to increase moderately due to climate change with rising temperature and CO_2_ levels and under a range of low to high precipitation scenarios, in particular for East and Central Africa. In the scenarios where cassava yields may be declining, this will be very moderate relative to many cereal crops [31].

Concerning pulses (grain legumes), in general, yields of common (*Phaseolus*-type) beans are predicted to experience yield reductions due to climate change. However, certain pulse crops are more resilient to hot and drought-prone climates, for example, cowpea, which is widely cultivated in SSA as a protein source [36]. Moreover, crop rotation of pulses such as cowpea with cereals like sorghum and pearl millet can substantially increase their yields [37]. Where studies on the effect of climate change on the productivity of peanuts are inconsistent, the productivity of Bambara groundnuts is even estimated to benefit from moderate climate change, depending on the variety [31].

Albeit with the uncertainty as to the overall impacts of climate change, especially in view of large regional variations, it can be projected that wheat, maize and common bean productivity in SSA will decline significantly, emphasizing the urgency to invest in CRCs productivity and use. Making a considerable shift to produce CRCs can allow SSA’s agri-productivity to keep pace with the needs of its growing population.

#### 2.2.2. Water

According to the FAO [38], the percentage of our planet affected by drought has doubled in the last 40 years. In SSA, more than 80 percent of the cropland is low-input rainfed production, and only three percent of the land is irrigated. As a consequence, crop productivity is highly affected by fluctuations in precipitation. A recent FAO report [6] states that, currently, about 50 million people in SSA live in areas where severe drought has catastrophic impacts on cropland and pastureland. It is also stated that up to 49,000 farmers in SSA lost their livelihoods due to recent droughts. On the positive side, a trend is observed of many small-scale farmers who are developing their own small-scale irrigation equipment, including buckets, watering cans and treadle pumps. These solutions tend to have lower unit costs and better performance relative to those managed by government agencies. The FAO report concludes that there is considerable potential to expand profitable small-scale irrigation in the region, but water saving and modernization of irrigation are required. CRCs can play an important role since they are adapted to abiotic stresses such as drought and higher temperatures [34].

#### 2.2.3. Soil and Minerals

Among CRCs, pulses are known for their ability to fix atmospheric nitrogen. Therefore, they reduce the amount of fertilizer required. For this reason, pulses can also improve the productivity of cereals by crop rotation [37]. Drought-tolerant CRCs are usually also deeper rooting compared to, e.g., maize, and hence stabilize soils and prevent soil degradation [39]. Probably related to this, there is evidence that sorghum has higher nitrogen uptake efficiency than maize when cultivated in the dry tropical savannah [40]. Therefore, in general, promoting CRCs will aid sustainable agriculture to prevent soil depletion and even reduce the need for new farmland and associated deforestation.

#### 2.2.4. Biodiversity

The predominant focus worldwide on the cultivation of a few staple food crops (in particular maize, wheat and rice) has evidentially reduced the heterogeneity and diversity of the arable land crops. In fact, over 50% of the proteins and calories in the global diet are provided by only these three crops [41], which cover 40% of all the arable land globally. Furthermore, of the 7000 edible plant species cultivated or harvested, only 150 crops are commercialized on a significant global scale [41]. This lack of diversity in the food production system negatively impacts the ecosystem and biodiversity directly and indirectly. In SSA, demands and expectations of modern supply chains have led farmers to concentrate on fewer and fewer crops, without consideration of indigenous knowledge and local communities, which has resulted in a steady loss of agrobiodiversity [42]. However, CRCs are strategic resources necessary for the well-being of millions of people, particularly of those living in marginal areas. It is expected that CRCs (and other local crops) will add spatial and temporal heterogeneity in cultivated crops and arable land, which will enhance the ecological resilience and biodiversity while broadening the food basket to meet the nutritional requirements of people in the region.

### 2.3. The Food Supply System of CRCs

#### 2.3.1. Agricultural Production

Historically, the SSA food system relied on a diversity of indigenous crops, with cereals, sorghum, millets and starchy roots and tubers as the primary staples [43,44]. As discussed in Section 2.2.3, SSA has seen an overall shift to maize and cassava as staple crops due to their high primary productivity. In the period 1960–2013, wheat imports grew at similar rate as urban population, with a 10-fold increase (Figure 2). On the contrary, sorghum and millets only showed a moderate increase in production in SSA (about a fivefold increase) despite the rapid population growth. This clearly indicates that these traditional crops are declining in their contribution to the diet. 

The growing demand for wheat has resulted in an increase in wheat production in SSA, but the increase in production has been negligible compared to the increase in wheat imports (Figure 2). As a result, the lack in self-sufficiency related to wheat production is being exacerbated. There is potential for increasing wheat production in some areas of SSA, like the Horn of Africa. However, humid tropical regions are unsuitable for wheat cultivation due to heat stress which reduces growth and yield, but they are very well-suited to cassava, hence the large increase in cassava production [45,46]. Wheat can be and is cultivated in the hot drier regions but they are better suited to cultivation of the tropical (C4) cereals maize, sorghum and millets because of their potential to produce higher yields as they can better utilize the high intensity solar energy in the tropics [47].

#### 2.3.2. Trade

The dependence of SSA on imported wheat continues to increase with, for example, Kenya and Uganda importing 68% and 95%, respectively, of their domestic needs. The increasing importation of staple crops negatively affects the SSA economies due to the cost in foreign exchange and the fact that it constrains the development of local agri-food value chains and economic activities.

Despite the increase in maize and cassava production in SSA, their food value chains, like those of CRCs, remain predominately traditional, being short with farmers often selling their produce directly to end consumers or small-scale traders and processors because smallholders face barriers to supply to process their produce as processed or convenience food [5]. Therefore, these crops currently provide very limited value-added processing activities and consequent economic benefits to local communities. Whereas wheat, soya and some maize are traded in the formal market channels, CRCs are mainly traded in local and informal markets and hence their (local) availability is sometimes limited. This effect is further aggravated since CRCs are often cultivated as subsistence crops. For mature businesses, a secure and reliable availability of a particular volume of crops with constant price and quality is essential to produce added-value ingredients or food products. The availability of adequate quantities of CRCs suitable for bread-type product applications and at an economically convenient price relative to wheat is considered as the main hurdle for increasing their use. This is especially the case for products aiming at lower-income consumers. Although the cost of crops varies locally and in time, general trends on availability and price can be derived by looking at the situation in some of the SSA countries, such as Kenya, Uganda and South Africa. Trends in availability and price in these countries are provided in Appendix A, respectively. Regarding costs, maize is generally the cheapest cereal and is considerably cheaper than wheat. Similarly, cassava flour is relatively inexpensive, especially in cassava-producing countries like Kenya and Uganda. All other CRC cereals are generally substantially more expensive than wheat. Nevertheless, some exceptions based on high local produce can be observed, as in the case of Uganda where sorghum flour is the cheapest cereal flour. Amaranth is the most expensive starchy grain and flour in all countries. All legumes are substantially more costly compared to wheat for all these countries. Among the legumes, *Phaseolus* beans and soya beans are the least expensive due to relatively high availability and productivity, respectively, but remain substantially more expensive than wheat. Cowpeas are similar in cost to superior *Phaseolus* beans in Kenya but far more expensive in Uganda. 

#### 2.3.3. Food Processing and Transformation

As stated, CRCs are mainly used for making traditional staple foods in SSA. Sorghum, maize and millets are traditionally used and account for a major part of the energy and protein intake [43,48,49]. The processing and transformation of these crops into staple foods is highly diverse as a result of local traditions and cultural identities. However, processing practices can, in general, be categorized into five main groups: (i) boiling and/or roasting, (ii) nixtamalization by commonly cooking with alkaline rock salt, (iii) sprouting and malting, (iv) milling or grinding and (v) mixed microflora fermentation with lactic acid bacteria and yeasts. While boiling and roasting are also used for direct consumption of the kernels, the other practices usually involve additional processing. Typical examples of staple foods that are popular in SSA and result from a combination of such processing practices are flatbreads (pancake-like), couscous, dumplings and porridges. The common features of these foods are that they are all compact, with a dense structure and not or very little aerated. 

Aerated wheat bread and bread-type products were introduced by European colonists and settlers [50]. Contrary to the traditional staple foods, these wheat-based products are generally characterized by an appealing spongy structure with a soft inner crumb-type structure and crispy–crunchy crusts after baking or heating them. Since their introduction, consumption of bread-type products based on refined wheat has become increasingly popular in SSA; also due to their convenience, they have progressively displaced traditional meals. Nowadays, breads and bread-type products available in SSA are quite differentiated and widely geographically dispersed across the region. A schematic overview of these products is provided in Figure 3.

The unique structure and texture of wheat-based breads and bread-type products is predominantly due to the ability of wheat flour to produce a workable viscoelastic dough when hydrated and kneaded [51]. Such a dough results from the formation of a protein network, the gluten, which aids in the incorporation and stabilization of air [52,53]. On the contrary, breadmaking with CRC flours necessitates the utilization of a liquid batter, which cannot be easily handled and has limited ability to hold gases [54]. These differences in the processability of wheat flour on the one hand and CRC flours on the other are the main reasons behind their distinct use in different types of staple food products. 

Traditional processing practices for CRCs have been used for the purpose of functionalizing these ingredients for bread-type applications. Most commonly, functionalization consists in precooking part of the flour to thicken the dough in order to hold some of the gasses produced during fermentation [55,56]. Additionally, the flour–water mixture typically undergoes a mixed lactic acid bacteria and yeast fermentation [56,57], which gives them a somewhat leavened texture and an acidic flavor. These sourdough preparations are labor-intensive and time-consuming as they generally take a few days. 

The traditional processing practices, aside from bringing functionality to CRCs with regard to texture and flavor, also have important nutritional and food safety purposes. Lactic fermentation improves protein and carbohydrate digestibility, increases B vitamins and minerals availability [56,57,58,59]. The reduction of pH below 4 by the fermentation also inhibits the growth of pathogenic bacteria [60]. The process of nixtamalization is also commonly used in Latin America as a means to produce a cohesive dough and flexible flatbread from maize and sorghum, reduce cooking time, improve digestibility and flavor, as well as for preservation [61]. From the nutritional perspective, nixtamalization improves niacin bioavailability, increases calcium intake and dietary fiber intake by increasing the content of resistant starch and reduces the level of phytic acid [61,62,63,64].

With increasing urbanization and expanding urban food markets, the demand for processed and prepared convenience foods has been increasing rapidly [64,65]. Street foods provide a substantial amount of the diet for a wide range of social groups in urban areas, notably the poorer people [66]. Furthermore, they are the least expensive and most accessible means of obtaining a nutritionally balanced meal outside the home. Among the types of street foods, cereal-based products are the predominant group [67]. The chapatti-type flatbreads of East Africa are convenience-type products, which are not traditionally from SSA and are mainly wheat-based, often produced by street vendors. Another increasingly popular bread-type product is based on deep fat-fried dough or batter. Examples are mandazi, magwinya and samosa [68]. These can be plain, sweetened or savory products with or without filling. Since they are based on a dough or batter which is fried, their production is simple and does not require expensive utensils, hence they are often prepared and sold on the street. Fried breads are almost without exception made from refined wheat flour providing an aerated soft texture with a crispy crust. Their high fat content, frequently added sugar and because they are made from refined wheat results in them having a high energy density but being rather poor in nutritional quality. Steamed breads and dumplings also have a component of convenience as people often take them to consume while traveling long distances. In West Africa, they are still largely made with cassava, yams and maize [43,69]. In Southern Africa, they were traditionally made with sorghum and pearl millet, but today, they are largely made with wheat flour [70]. The Southern African traditional baked breads are today mostly made with wheat flour but can be composited with other grains or flour, most commonly maize [70].

As is evident, wheat largely dominates the production of bakery and bread-type products in SSA. Several approaches have been studied to create aerated bakery products based on alternative, non-wheat flours [51]. Those include the use of additives such as hydrocolloids as well as the modification of flours through bioprocessing technologies such as fermentation or physical treatments such as heat. With respect to flour modification via processing, African indigenous knowledge could be incorporated and revisited in the use of CRCs for attractive modern food products. To be successful, these approaches should exploit crossovers in the diversity of traditional practices to limit additional processing costs and complexity in order to ensure commercial competitiveness compared to imported wheat.

#### 2.3.4. Food Market and Consumption

Considering the heterogeneity of the SSA consumer market as described in Section 2.1.2, we focused here on the most important and most rapidly growing market: the evolving urban market, relatively informal, with low-income to low-middle class, relatively young consumers. For these consumers, the trends and the demanded attributes for bread-type products can be largely related to affordability, convenience, Westernization and health awareness. 

*Affordability.* In SSA, like elsewhere, the needs and wants of consumers at different socioeconomic levels are different. Socioeconomic restrictions, especially at the lower end of the market, will at times override preferences [71]. The SSA situation is probably similar to that found in a study conducted in Uruguay where low socioeconomic class respondents described their choices as mainly driven by economic factors and physical needs (e.g., satiety), whereas product-related characteristics (i.e., convenience) are more important for middle socioeconomic class respondents [72]. However, it should be noted that poor consumers are highly concerned about food quality and value as they apply a much larger proportion of disposable income to food compared to more affluent consumers [73]. They simply cannot afford to make purchase mistakes. Economically deprived consumers are quality-conscious and aspire to consume products of high value. The perception of quality, however, lies in the hands of the beholder.

*Convenience.* The convenience trend continues to be a major factor in the food industry as most consumers have busy lifestyles with little time to spend on food purchasing, cooking and eating. It has been highlighted that access and availability have a significant effect on daily consumer consumption and dietary intake in developing countries [74]. Socially home-prepared meals are losing their significance while convenience products, fast food and restaurant meals are gaining in importance. Specifically concerning SSA, the demand for convenience largely results from changes in social habits; increased working hours, especially for women, and changing household structures [75]. Urban consumers indicate that they have very little time to cook, shop or prepare food and therefore prioritize products and services that assist and/or replace personal effort [76]. It is therefore not surprising that in South Africa, for example, out-of-house consumption of food accounts for 36% of the total food expenditure, a significant and growing proportion of the total food intake [77]. For bread products, important convenience aspects include ease of availability (e.g., street vendors near working places), consumption on the go, no need for further preparation and versatility to be consumed on its own or combined with other meal components.

*Westernization*. Younger consumers, in particular, in many African countries aspire to consume food types of the modern world [78]; types that are commonly featured on social media. Access to and consumption of certain food types (e.g., Western-style products) can serve symbolically as escape options from difficult situations and poverty. Among children and teenagers, avoiding “low-status” traditional food may be a way of avoiding stigma in an effort to depict themselves as contemporary. This behavior was described through observation of a marginalized ethnic group in Argentina living under the burden of negative societal attitudes [79]. For CRC-based products, this trend can be a potential threat as food products based on traditional grains and pulses are often viewed as “the antithesis of cool”, especially amongst younger consumer groups. This is especially the case in South Africa where younger consumer groups often frown upon more traditional food products and often refer to it as “poor man’s” food [44,80]. Important to note here is that information about consumers’ perception of gluten-free bread or breads from CRC flours in SSA is extremely limited. The research focus is often on the supply side while information related to the demand side is neglected [81]. 

*Health awareness*. Consumers globally are becoming more aware of the impact of food choices on health. Well-being and healthy lifestyles are becoming megatrends, especially amongst higher-income, more educated consumer groups [82]. Despite this, statistics from the 2019 Global Nutrition Report indicate that the youth from emerging economies such as SSA still consume high quantities of unhealthy foods and beverages, thus exacerbating the problem of poor nutritional status [83]. In South Africa, data pertaining to bakery items indicate that even though the health trend is emerging, consumption of unhealthier processed staple foods such as white bread and highly refined maize products are often still preferred compared to the healthier options [20,75]. Table 2 presents a summary of the diverse set of factors that consumers from different SSA countries consider when evaluating choices of bread products. The products should provide elements of value that address a complex range of functional and emotional needs and desires of the consumers (e.g., affordable, well-tasting, nutritious and convenient). The sensory quality and perceived consumer value of food items contributes much to the emotional well-being of consumers [84] and often drives product acceptance more than nutritional quality [6,85]. The food environment in which people live plays a large role in their diets. As explained, urban consumers tend to be more convenience-driven compared to their rural counterparts. For lower-income groups, which predominate in SSA, consumers might have more time but lack even basic food preparation equipment. Therefore, food products need to present characteristics that ease the burden of home preparation, i.e., inexpensive fast foods with a high satiety value. Based on studies in Kenya and Burkina Faso, it was concluded that traditional grains (sorghum, millets and maize) need more complex preparation, and in those markets, wheat product consumption (e.g., bread) reflects ”a preference for convenience rather than a taste for wheat per se” [86]. Lack of awareness and understanding of product benefits and unique attributes is also considered a barrier to acceptance [81]. For example, it has been found that consumers in Ghana considered safety, followed by taste, packaging, labeling information, texture, aroma and color as being important attributes of cassava–wheat composite bread [81]. Overall, the study identified important positive price-related and health-related consumer perceptions related to cassava–wheat composite bread compared to pure wheat bread. 

In a study to determine Kenyan consumer preferences for wheat buns with added insect flour, it was found that respondents could be classified into two groups [87]. The first group had a “tendency to try new food products” having a strong promotion-oriented food choice motive, and this tendency increased with age, while the other group tended to prefer known (and safe) products displaying prevention-orientated food choice motives. Insights into the motives and barriers that consumers experience are critical to identify the opportunities and threats related to the introduction or promotion of new food products.

Despite the fact that health and nutrition are increasingly playing a role in the decision-making process of middle- to high-income consumers, many South African consumers, and presumably in other SSA countries such as Malawi [105], are price-sensitive and prefer lower-priced breads [75]. Hence, for CRC-based products to be successful, cognizance needs to be taken of consumer preferences such as higher-income consumers placing a greater emphasis on quality, nutrition and sensory characteristics compared to lower-income consumers who often forgo these characteristics for affordable prices. 

To motivate consumers to make healthier food choices, African indigenous knowledge could play an important role. Indigenous knowledge and practices include the collective knowledge, skills and technology of the local community and their environment [44]. Tapping into the appreciation of the traditional food culture in modern CRC-based products could increase their appeal.

### 2.4. Outcomes of the Food System

#### 2.4.1. Food and Nutrition Security 

Climate change including higher temperatures and CO_2_ levels, changes in rainfall patterns and more frequent extreme weather events such as droughts and flooding will impact food security in SSA severely. Whereas the climate change in moderate climates will even allow increased agricultural productivity, the tropical and arid regions of SSA are facing large disadvantages. The overall expected effects of the predicted climate change are to reduce the productivity of wheat, maize and common beans in SSA by 18–35% by 2050 [31]. Hence, without switching to more CRCs, climate change will cause reduced agricultural productivity in SSA and compromise food security. In particular, the poorest and rural populations are most vulnerable. 

CRCs, in particular sorghum, millets, amaranth, cassava, cowpea and Bambara groundnuts are predicted to sustain their productivity on average or even show some increase in productivity [31]. In addition, in most cases, less water is needed for their cultivation. By investing in improving the yield and use of these crops, a flourishing CRC food production chain will not only contribute to food security, but also provide employment and income, hence contributing to the local communities’ ability to secure their primary food demands and increase their disposable incomes. Furthermore, the resultant reduced dependence on wheat importation and consequent reduced price volatility of crops and food prices will contribute to secure food affordability and availability for the poorest.

#### 2.4.2. Safe and Healthy Diets

Food safety is not primarily affected by the type of crops selected for making breads and bread-type products. Independent of their climate resilience, all the various cereals, roots and pulses reviewed in this paper may contain certain undesirable components or risks. These components can consist of various types of antinutritional factors, allergens and toxins (see Appendix A). Prevention of safe intake levels of these components being exceeded requires attention across the entire food supply chain to ensure food safety so as not to favor utilization of a certain crop. The main exception is the toxic cyanogenic glycosides present in cassava, particularly in bitter varieties, which require specific processing operations to prevent acute toxicity and chronic exposure. The antinutritional factor phytate, which is present in all cereals and pulses, is a major concern as it contributes to malnutrition by decreasing mineral absorption. These compounds can be reduced by certain food processes like fermentation, underlining that processed CRC-based foods can contribute to a safe diet. 

Healthy diets consist of a variety of unprocessed or processed foods balanced across the food groups and consisting largely of plant-based foods: wholegrains, legumes, nuts and an abundance and variety of fruits and vegetables, with moderate amounts of animal products: eggs, dairy, poultry and fish; and particular small amounts of red meat [106]. Concerning improving dietary diversity, as indicated, over 50% of the proteins and calories in the global diet are currently provided by just three crops—maize, wheat and rice [106]. The authors of this report state that “Over-dependence on a few plant species exacerbates many acute difficulties faced by communities in the areas of food security, nutrition, health, ecosystem sustainability and cultural identity”. 

Healthy diets are not only affected by the type of the staple crop consumed, but also by the level of refinement of the foodstuff. A comprehensive overview of the nutritional composition of the different staples, as well as some of their flours is provided in Appendix A, respectively. In general, the tuber and root crops are predominantly starch sources and provide very limited protein and minerals like iron and zinc to the diet. Cereals are specifically richer in proteins, but are generally low in the essential amino acid lysine. The pseudocereal amaranth provides a substantially higher protein and lysine content. Pulses are of specific interest in that cereals and legumes complement each other in their protein content and amino acid composition. Where cereals are limited in the essential amino acid lysine, pulses are rich in lysine. Hence, producing food products based on combinations of cereals and pulses will provide the desired balance of essential amino acids for the human diet. It should be specifically noted that the refinement process can severely reduce the nutritional quality. For example, in the case of wheat, refined wheat flour has only 25% of the fiber content of wholegrain flour and, more importantly, of all the associated vitamins and minerals as these are concentrated in the germ and bran layers [107]. Additionally, the lysine content is further reduced. The same holds for all cereal and pseudocereal CRCs, emphasizing that the use of CRCs should be accompanied by food processing limiting refinement, combining the variation in nutritional values of diverse crops for producing attractive food products. 

People with low income are the most food insecure and in general their diet is also the least diverse [108,109,110]. For example, in SSA, some populations rely on maize or cassava as their essentially only staple. In Lesotho, for instance, the average maize consumption is 328 g maize per person per day for the country’s entire population compared to an average consumption across SSA of 141 g/day [111].

As stated, in SSA, the lower middle class with daily per capita expenditures of $2–10 is the fastest growing and the most rapidly urbanizing group. With growing disposable income, urbanization and women’s participation in formal employment, this group rapidly abandons their traditional meals based on indigenous crops and instead increases their consumption of convenience-type staple foods such as refined wheat breads. Therefore, their increased disposable income often leads to a less healthy diet and increased risk of micronutrient deficiency. As these products are often rich in starch and fat and low in proteins and dietary fiber, this change in lifestyle adds to risks of overnutrition and obesity, which can coexist with undernourishment and micronutrient deficiencies. Introducing CRCs like sorghum, finger millet, amaranth and, in particular, pulses into convenient staple foods like bread would substantially improve nutritional quality of their diet. 

The upper middle class would be expected to switch from starchy plant-dominated diets to more varied foods that include a broader range of fruit and vegetables and animal-sourced proteins, and hence improve the nutritional quality of their diet [108]. To some extent, this may be true, but the increasing prevalence of noncommunicable diseases (NCDs) including obesity indicates that increasing disposable income may contribute to a higher consumption of highly processed and convenience products instead [112]. Hence, provision of modern and convenient staple foods like bread with a high nutritional quality by including CRCs would also substantially improve their diet.

From the above, it can be stated that partially replacing starch-rich staples like refined wheat, rice, maize and cassava with CRCs will provide a more nutritious diet rich in proteins, dietary fiber, essential fatty acids, vitamins, minerals and phytochemicals. This is especially where these CRCs are only refined to a limited extent. Hence, replacing refined wheat flour with these CRCs can contribute to addressing all forms of malnutrition (the triple burden of undernutrition, micronutrient deficiency, overweight and obesity) and reducing the risk of diet-related NCDs.

#### 2.4.3. Inclusiveness and Equal Benefits

Increasing the utilization of CRCs in bread ingredients and bread-type products in SSA offers many more business opportunities along its value chain compared to their refined wheat counterpart. In particular, local and regional food systems could be enriched with value-adding activities to stimulate local community economic development. Such economic activities by local communities grouping together would enhance their collective efficacy and empower them, enhancing community capacity [113]. Sharing of resources contributes towards food utilization, food access, food availability and, ultimately, food security [114].

#### 2.4.4. Sustainability and Resilience of the Food System

CRCs are notably resilient to abiotic stresses such as limited water availability and extreme temperatures [33]. Hence, expanding the cultivation and utilization of CRCs in SSA will contribute to more secure agronomic production in the face of climate changes. Additionally, utilizing these CRCs as ingredients for processed foods such as bread-type products instead of relying on imported wheat will enhance the resilience of the entire SSA food system. However, a clear prerequisite is that the productivity of CRCs needs to be improved to assure that enough food can be produced on the available arable land to feed the fast-growing SSA population.

## 3. Proposed Interventions from the Food System Approach Perspective

Notwithstanding the advantages of CRCs to SSA with respect to their production, use and consumption, their potential is still highly constrained. In fact, with SSA consumers increasingly abandoning traditional diets and adopting Westernized diets, the intake of CRCs is even declining. The food system approach applied in this paper provides a comprehensive framework to gain systematic insights in the opportunities, challenges and hurdles to overcome for valorization of CRC crops in SSA for bread-type products. Based on the outcomes of this analysis, we propose that interventions are required throughout the food system, which together could facilitate and drive this shift from wheat to CRCs.

**Investment in CRC-specific agriculture in SSA is required to improve yields and productivity to make it more attractive for farmers to produce these crops and increase their availability**. Much is already being done in this direction, such as the development of Africa’s seed systems [115]. Here, the potential of two emerging technologies, precision agriculture and genome editing, will be highlighted. Increasing crop yield and productivity are brought about by two factors: improved plant genetics and agronomic practices. Up until now, both have primarily benefitted large-scale agriculture due to their high development and implementation costs. The need is now to additionally empower smallholder commercial farmers.

*Precision agriculture* can be defined as the application of new technologies to increase crop yields while at the same time reducing the levels of the inputs needed to grow crops (land, water, fertilizer and biocides). The rapid advances taking place in information technology, notably harnessing the power of mobile phones, is making precision agriculture affordable to smallholder farmers. An example of an intervention is the optimization of a farm’s sorghum crop design (especially plant density) for each season. Using crop modeling and seasonal climate forecasting, farming profitability could be increased by up to 21% even on low-potential soils [116].

*Genome editing* is now being applied to several CRCs. For example, CRISPR/Cas9-mediated editing of the cassava genome has been shown to improve resistance and reduce the severity of infection by the cassava brown streak virus [117]. This pathogen is estimated to cause a $175 million loss in cassava yield in East Africa. Genome editing has two important advantages compared to recombinant DNA technology (genetic engineering). It is a much simpler technology and hence considerably less expensive, and because it does not involve the insertion of foreign DNA, obtaining regulatory approval for the release of new varieties is far less complicated.

As the required scale of agricultural research and development investment is very high, significant impact can only be achieved through the *transnational regional government organizations* in SSA, for example, the Southern African Development Community (SADC) and the East African Community (EAC). Moreover, international governmental and nongovernmental development agencies such as ICRISAT (the International Crops Research Institute for the Semi-Arid Tropics) and AGRA (the Alliance for a Green Revolution in Africa) and private agri-business sector activities must be fully integrated into the programs.

Lastly, farmers will not adopt climate-resilient crops solely on the basis of their climate resilience [118]. Besides access to and availability of climate-resilient crop varieties with good yield, farmers need *education and extension services*, training and support in techniques such as soil treatments, fertilization, planting dates, etc. Additionally, crop-breeding programs must consider farmer and market trait preferences.

**Investment in processing technology is required to improve the functionality of CRCs for making a variety of healthy foods**. In particular, functionalities are required for manufacturing processed food products at the industrial scale. Functional food ingredients based on CRCs are an important step towards maturing their value chains and will provide opportunities for value-added exports as well as value addition within the SSA domestic food industry. Importantly, the inclusion of CRCs in staple foods and improvements in the nutritional quality of street food should not be at the expense of higher food prices as that will put them out of reach for a large segment of the urban population. From this perspective, traditional practices should be carefully revised and modernized by implementing affordable and efficient technologies. Consumer demand for clean label and mildly processed sustainable foods in the Western countries has resulted in considerable “modernization” of ancient technologies such as sourdough and yeast fermentation. Lactic acid bacteria fermentation provides an affordable way to modify the main food structure-building components such as starch, non-starch polysaccharides and proteins, leading to improved functionality [119]. Novel application of this traditional technology requires optimal choices in the proportion of sourdough to be added, in fermentation conditions and in the use of appropriate (indigenous) starter cultures [120,121]. Additionally, interesting novel food products and ingredients can be produced through the concept of crossover fermentation [122], where microorganisms from a traditional environment are added into a new environment to obtain a novel food product, which can be altered by the use of traditional practices. Such novel approaches combined with the enormous diversity of microorganisms used in traditional fermentation processes increase the opportunities for functionalization of CRC flours.

*Street foods* have received insufficient attention as a valuable source of macro- and micronutrients, and their potential role in achieving nutrition security has been neglected. Most vendors do not sell enough variety of foods to provide a healthy diet [67]. Considering the growing importance of street food in the context of urbanization and provision of nutritious foods across different segments of the urban population, acting on processing and formulation of CRCs into street food provides large opportunities to improve food and nutrition security and healthy sustainable diets. These interventions can comprise (i) practical guidance in formulating street food with CRCs based on scientific insights on CRC functionality, (ii) revisiting traditional practices for improved functionality of CRC ingredients in street food products and (iii) development of new healthy and convenient foods, which address nutritional needs and local consumer demands.

*Natural biofortification* could be addressed in processing of CRC for functionalization. This would enable nutritionists to concomitantly address micronutrient deficiencies in staple foods while avoiding the costs of extrinsic fortification. Controlled sprouting of CRCs can be a potential way to functionalize them for bread-type applications [123,124], while also enhancing the bioavailability of minerals and vitamins. Under the sprouting conditions optimal for phytase action, the bioaccessibility of minerals such as Zn and Fe can be increased in cereal grains such as sorghum and millets by up to 30%, but potential health benefits should be further investigated [124,125]. Additionally, sprouting enhances the vitamin content of grains and the relatively high levels of vitamins E, B and C in sprouted cereals can contribute significantly to daily intakes of these vitamins. Fermentation of cereal, pseudocereal and legume materials with selected bacterial strains is also an effective means of biofortification of CRC-based staple foods to address vitamin B12 deficiency resulting from limited consumption of animal-derived food [126,127]. 

**Consumer-centric development of attractive products containing CRCs**. Within the SSA context, the complexity of the fragmented consumer market with strongly diversified consumer segments as well as the huge range of local products need to be taken into consideration. It is proposed to develop customizable products and marketing concepts that can be effectively diversified and adapted to the needs and demands of the local market. 

**Raising consumer awareness and understanding of the positive aspects of CRCs**. Their nutritional quality and health aspects, sustainable aspects and cultural value/heritage need to be promoted to counter the current misconception that CRCs are merely “food security crops” and CRC products are a “poor man’s” food.

**Creation and empowerment of short food supply chains in urban environments**. Short supply chains should be promoted, in which the use of effective and cost-saving processing of CRCs for staple food production is supported by providing a safe water supply and appropriate processing technologies to street food vendors. This will additionally stimulate employment in local communities.

**Economic incentives and policies are required to promote CRCs throughout the supply chain**. Interventions at the agricultural, food technology and consumer levels should be supported and enabled through national and transnational polices and incentives. This requires integrated programs involving multiple stakeholders to stimulate CRCs, their trade, processing and consumption and at the same time taking into account the challenges described in Table 1. All the above interventions, guided by the holistic food system approach, need to be well-orchestrated to create a simultaneous pull from the consumer and the market and push from all the stakeholders in the food supply chain back to the policymakers responsible for land use planning in agriculture. 

## 4. Conclusions

The rapid transition in diets and lifestyle in SSA resulting from massive urbanization and increasing disposable incomes makes its population increasingly vulnerable to the double burden of malnutrition and obesity. The situation is exacerbated by the increasing pressure to produce sufficient food and provide employment for this growing population and by the predicted negative effect on crop yields resulting from climate change. Despite the urgency and the interventions made in addressing sustainable solutions (such as increasing cultivation of CRCs and their utilization in staple foods, i.e., bread), several programs have had limited impact. In this research, we applied a food system approach to the bread food value chain in SSA to identify the key success factors as well as challenges that need to be addressed to bring about the widespread use of CRCs in bread in the region. The food system analysis provided a systematic insight in the challenges and hurdles that need to be overcome to increase the uptake of CRCs and indicated that the following interventions are needed to achieve success: Investment in CRC-specific agriculture in SSA to improve yields and productivity to make it more attractive for farmers to produce these crops and thereby increase their availability;Investment in processing technologies to improve the technical functionality and fortification potentials of CRCs for making a variety of healthy and attractive foods;Consumer-centric development of CRC-enriched attractive and convenient food products within the context of urbanization;Raising consumer awareness and understanding of the positive attributes of CRCs, including the importance of preserving indigenous knowledge and culture;Creation and empowerment of short food supply chains in urban environments;Economic incentives and policies to promote CRCs throughout the supply chain through integrated programs involving multiple stakeholders.

Achieving a substantial increase in the cultivation and consumption of CRCs should lead to considerable nutritional, economical and sustainability improvements in SSA.

## Figures and Tables

**Figure 1 foods-11-00135-f001:**
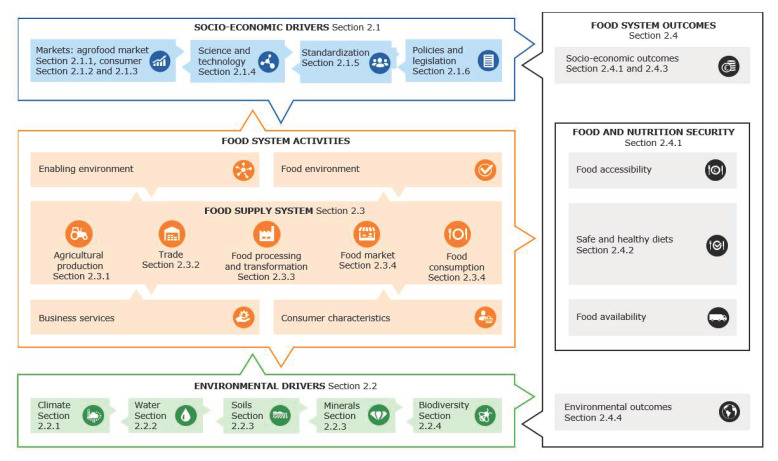
Food system analysis of the bread food value chain in sub-Saharan Africa, modified with permission based on [3]. The paragraph numbers included refer to the sections in this paper.

**Figure 3 foods-11-00135-f003:**
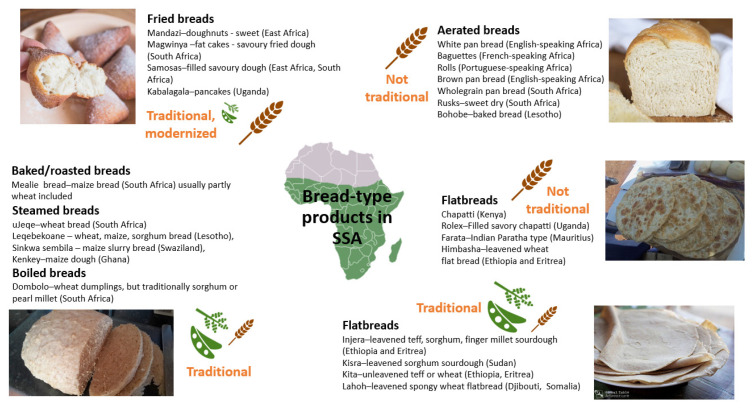
Bread and bread-type products in SSA. Overview of the main product categorization with nonextensive examples of products and the country/region of origin. For each product group, an indication is given whether or not the product can be considered traditional, and the size of the icons for wheat [
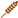
] and CRC [
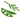
] indicate ingredient use.

**Table 1 foods-11-00135-t001:** Reasons proposed for the lack of success and limited impact of wheat flour replacement programs.

Reasons	FAO Composite Flour Programme [24,25]: Worldwide	National Composite Flour Programs Arising from the FAO Programme [26]: Bolivia, Brazil, Colombia, Senegal and Sudan	Cassava Flour Inclusion Policy [27,28,29]: Nigeria
Policy	- Regular supply of raw materials of consistent quality and stable price is required - Structural policies are needed for the production and pricing of agricultural commodities - Findings made by African research are not taken up quickly by the industry - Infrastructure is needed to link research to industry	- Underestimation of the complexity of a national composite flour program and consequent underallocation of management expertise and finance - Lack of subsidies of motivate the various players in the food chain	- Frequent government policy changes with respect to the required level of cassava flour inclusion - Inadequate legislation and implementation of the legislation to facilitate successful implementation of cassava flour inclusion policies
Regulatory	Grain standards required	- Lack of regulatory methodologies and systems - Simple methods not available to monitor levels of non-wheat flours in wheat blends	
Food processors	- Training of personnel is vital to ensure appropriate application of the technologies - The great variability in sorghum and millet and high protein (pulse) grain quality causes problems for millers	Lack of incentives for millers and bakers to participate in the program	- Flour millers unwilling to comply with governmental policy - Necessary training and technologies not provided to bakers
Processing and food quality	- Development of suitable small-scale processing equipment needs to be prioritized - Setting up of pilot plants needed to avoid failure at the industrial level - Considerable variations in bakery product processing quality of sorghum and millet flours - Problems with starchy and pulse flour grittiness and cohesion. Flour pre-treatments required to eliminate these - Combination flour treatments such as pre-gelatinization, inclusion of gums and surfactants need to be studied improved baking performance - Research into application of composite flours in indigenous wheat bread types required, e.g., Arabic and Indian - Far more research into indigenous foods needed, e.g., to simplify their production and improve their keeping quality	- Impaired baking quality of composite flours is problematic for many bakeries - Small bakeries in particular lack the required equipment and processing aids and do not have funds to purchase them	- Cassava-wheat composite bread quality defects - Objectionable odor and color, and more rapid staling
Food safety	Potential issues of toxicology and food safety must be addressed		Potential presence of toxic compounds (cyanide) in cassava flour
Consumer	- Need to determine what sorts of bread the consumers desire, e.g., high-volume or firmer crumb types - Assays required to identify composite flours in order to protect consumer interests	Consumer hostility if the quality of traditional bread products is reduced or prices are increased	- Benefits of cassava-wheat composite bread not generally known - Consumers concerned about the possible presence of toxic compounds (cyanide) - Consumers believe that the bread causes bulkiness (feeling of bloating) - Strong consumer preference for 100% wheat flour bread

**Table 2 foods-11-00135-t002:** Examples of factors and characteristics that consumers in various sub-Saharan African countries consider when judging the value of bread.

Factor	Characteristics	Description of Realization (Examples)
Cost/ affordability	Value for money	Bread is the most basic foodstuff and should be relatively cheap–South Africa [88] In-store baked breads are cheaper than company-branded bread and hence more accessible to the poor–South Africa [75] Bread is made with imported flour and is therefore expensive–Malawi [89] The baking industry is very sensitive to exchange rates and international prices when a country is a net importer of wheat–South Africa [90] Bread as a staple should be affordable, have acceptable sensory characteristics and provide adequate nutrition–Nigeria [91] For low-income and very low-income consumers, there is (however) little leeway between choosing quality and accepting what price dictates [71] Consumers who are aware of cassava-blended flour bread and who like its taste and texture are willing to pay more than consumers who are unaware–Ghana [81]
Sensory properties	Visual	The appearance of bread is the most basic characteristic of a bread type and includes e.g., familiar shape and form, size, patterns and crust and crumb color, presence of flour or inclusions like seeds on the outer parts, shine–Nigeria [92] Visual appearance is an indicator of quality or suitability to consume, e.g., no mold growth—Nigeria [93] Lighter color bread is preferred-Malawi [89]
	Aroma/smell	The aroma/smell of bread is a basic characteristic of a bread type and a quality indicator–Ethiopia [94] The smell of freshly baked bread is appetite enhancing. “The smell of fresh bread makes me hungry”–South Africa [95]
	Texture/touch	The surface texture (e.g., smooth, rough) and crumb texture (e.g., pore characteristics and distribution) are characteristic of a bread type, e.g., injera (fermented flatbread) has a spongy texture–Ethiopia [55] Fresh injera should not break when rolled, i.e., is pliable–Ethiopia [55] Consumers are used to eating wheat bread that is porous and airy–Malawi [89] The inclusion of bran and coarse milled flour particles contributes grittiness and roughness, a dry feeling and after-swallow residual particles in the mouth–South Africa [68]
	Taste/Flavor	The flavor is a basic characteristic of a bread type and provides a lasting impression–Nigeria [96]. Bread flavor is relatively bland, but some types have a slight to strong sour taste (e.g., as a result of fermentation) or a flavor characteristic of the base material (e.g., cereal, root crop or legume flour)–Kenya [44] or with added tamarind–Uganda. [97,98] Old people like sour tasting bread more than the young generation, because younger participants are less familiar with maize and sorghum breads than they are with wheat breads–Lesotho [70] The bread or staple is considered an accompaniment and not the flavor hero of the meal, merely the bulking agent-Kenya [44]
Nutrition and health	Nutritional benefits	It must provide good nutrition for the family. Bread is an important staple and suitable for nutrient enrichment/ fortification-Malawi [89] It should provide energy to enable work, play and travel. It stills hunger pains, it fills/satiates. Some types help sustain energy levels for longer–Kenya [44] The types with fiber prevent or treat constipation–Kenya [44]
	Health and safety	It should be free from “perceived to be harmful-to-health ingredients”—South Africa [95] It should not lead to an undesirable prebiotic action, e.g., bloated feeling–South Africa [95] Some persons cannot tolerate certain ingredients/food components, e.g., gluten, allergens (e.g., wheat, soy, nuts) or exhibit negative reactions after consumption–South Africa [95] Persons with celiac disease cannot tolerate gluten–South Africa [95] Safety of blended cassava-wheat bread products must be given attention due to fears related to the consumption of cassava–Ghana [81]
Convenience and versatility	Convenience	Bread should be easy to carry, use, store and discard–South Africa [95] Bread is suitable for use while traveling/commuting/ for field trips, to take to work/school–South Africa [99] Saves time and energy–ready-to-eat, requires minimal meal preparation time–South Africa [95] Limited time available for shopping and meal preparation–Nigeria [78]
	Versatility—it can be used in many different ways	Pieces of injera are used as handheld utensils to pick up or contain the desirable component of a meal (e.g., vegetable and meat sauce)—Ethiopia [55] As a base for spreads/wrap/holder for fillings. It forms an important part of street food, examples of colloquial names: bunny chow, Gatsby, kota/spatlo and Sly—South Africa. These comprise a white bread half loaf filled with different savory foods, e.g., potato fries, processed meat, curry, sauces—South Africa [99] Bread products serve as breakfast for many, e.g., fat cakes (deep-fried batter) with tea—South Africa [99]
	Shelf life—how long it remains fresh and edible	Spoilage—the number of days that the product can keep until it is considered unacceptable from the microbial spoilage perspective—Nigeria [100] Staling—the number of days that the product can be kept until it is considered unacceptably stale—South Africa [101] Tamarind pulp is added during preparation of millet bread to preserve it for several weeks—Uganda [98]
Wellbeing	Religion	Bread is celebrated in many religions and beliefs, e.g., in Christianity—”give us this day our daily bread”. It is used in different forms during ceremonies, such as Holy Communion/Eucharist—a Christian rite or sacrament in most churches and an ordinance in others. Bread in Islam refers to food in general. It is a gift of God from the Creator to creation.
	Status	The bread type that is chosen to be consumed is an indication of social status–South Africa [102] The bread brand eaten is also an indication of social status. Consumers refer to trusted brands, familiar brands or those reminiscent of one’s upbringing–South Africa [103]
	Food security/sustainability/ ethical concerns	If there is no bread in the house, there is no food—South Africa [99] To our knowledge, no study with consumers from African countries have focused on concerns related to ethical or sustainability factors in relation to bread or bakery products. However, it may be fair to assume that a subsector of the population is increasingly aware that sustainable consumption is important to protect the natural environment, limit climate change and provide for social and inter-generational justice [104]
	Test of culinary skills	Bread making is a valued skill. For example, the skill of preparing bread has traditionally been considered as an indication of social upbringing of young women and their ability to handle the challenges of marriage and family responsibilities—Lesotho [70]
	Symbols of ethnic identity	The Acholi are known for their love of millet bread (dumplings) (kwon kal), and millet bread creates a sense of connection/inclusion and stability among this group in northern Uganda. “We are strong because of our millet bread”—Acholi, Northern Uganda [97] The Lugbara are a Central Sudanic people who are known for their delicious cassava bread (dumplings) (Inya)—Uganda [97]
	Celebrations/functions	Millet bread (dumpling) is eaten during important celebration functions e.g., “in honoring of twins, funeral rites, marriage and offering of sacrifices to the gods, people must eat millet bread.”–Uganda [97] Bread has always played a major role in celebrations–Lesotho [18]

## Data Availability

Not applicable.

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
