# Peer review of "Towards Sustainable Shifts to Healthy Diets and Food Security in Sub-Saharan Africa with Climate-Resilient Crops in Bread-Type Products: A Food System Analysis"

_foods, 2022, doi:10.3390/foods11020135_

Round 1
Reviewer 1 Report
Towards sustainable shifts to healthy diets and food security in 2 sub-Saharan Africa with climate-resilient crops in bread-type 3 products: a food system analysis.
Noort et al.
This is a very thorough and well-prepared article, describing the food system in Sub-Saharan Africa in relation to breads and alternatives. However, it is rather long and there is potential for some material to be reduced. For example, I’m not certain that Table 1 is required in the Introduction. The point could perhaps be more simply stated that traditional crops have greater heat and drought tolerance. The authors might explore other opportunities to reduce the overall length of Section 2.
In Table 2, “lack of government realism” and “lack of enabling policy environment” are too vague to count as reasons.
Sub-section 2.2.1. This subsection presents a very negative outlook for grain production in Africa based on model predictions. However, this is not consistent with the increasing production of all crops shown in Fig.2.
Line 803: “It is proposed to develop customizable product and marketing concepts that can be effectively diversified and adapted to the needs and demands of the local market.” Would it be possible to give some examples of what is in mind? This proposal is a little too vague in its present form.
Ditto for “Raising consumer awareness and understanding of the positive aspects of CRCs” and “Creation and empowerment of short food supply chains in urban environments.” Can you please elaborate.
Line 828: “and by the declining crops yields resulting from climate change.” As noted above, the data presented in Fig 2 shows increasing output of crops.
Some minor editorial improvement needed, e.g.:
Line 24 “pro-vided” (hyphen not needed)
Table 2 appears before it is introduced in the text.
Line 351 “CRC crops” If CRC refers to Climate Resilient Crops, then the second “crops” is not needed
Line 709 “The food systems approached applied in this paper” (approach rather than approached?)
Line 887. “prof. Yusuf Byaruhanga and dr. Dorothy Nakimbugwe.” Capitalize Dr and Prof
An overall proofread is warranted
Author Response
Thank you for this peer-review. We used all your advice to improve our manuscript. Please find in red our reply to each of your points in black.
This is a very thorough and well-prepared article, describing the food system in Sub-Saharan Africa in relation to breads and alternatives. However, it is rather long and there is potential for some material to be reduced. For example, I’m not certain that Table 1 is required in the Introduction. The point could perhaps be more simply stated that traditional crops have greater heat and drought tolerance. The authors might explore other opportunities to reduce the overall length of Section 2. We shortened the text of section 2 considerably and removed table 1 to supplementary table to improve readability.
In Table 2, “lack of government realism” and “lack of enabling policy environment” are too vague to count as reasons. “Lack of government realism” – Comment accepted and statement removed as the point is adequately covered in the statement “Underestimation of the complexity of a national composite flour programme and consequent under-allocation of management expertise and finance” “lack of enabling policy environment” – Comment accepted and the statement has been expanded to read “Inadequate legislation and implementation of legislation to facilitate successful implementation of the cassava flour inclusion policies”.
Sub-section 2.2.1. This subsection presents a very negative outlook for grain production in Africa based on model predictions. However, this is not consistent with the increasing production of all crops shown in Fig.2. The key message of this sub section is that wheat, maize and common bean productivity in SSA will decline significantly, emphasizing the urgency to invest in CRCs productivity and use. We added “Making a considerable shift to produce CRCs can allow SSAs agro-productivity to keep pace with the needs of its growing population.”
Line 803: “It is proposed to develop customizable product and marketing concepts that can be effectively diversified and adapted to the needs and demands of the local market.” Would it be possible to give some examples of what is in mind? This proposal is a little too vague in its present form. Ditto for “Raising consumer awareness and understanding of the positive aspects of CRCs” and “Creation and empowerment of short food supply chains in urban environments.” Can you please elaborate. The background and details of these recommendations is already elaborated on in section 2, and is at the disposal of readers who are interested in the consumer and marketing aspects. We decided not to elaborate further in section 3 to not further increase the length of the paper
Line 828: “and by the declining crops yields resulting from climate change.” As noted above, the data presented in Fig 2 shows increasing output of crops. The historic results of intensification and expansion of cultivated land can’t continue. Climate changes are predicted to at least depress the growth in productivity. The food production in SSA can’t keep up with population growth. We revised the text to better express: “predicted negative effect on crops yields resulting from climate changes.”
Some minor editorial improvement needed, e.g.:
Line 24 “pro-vided” (hyphen not needed) corrected
Table 2 appears before it is introduced in the text. corrected
Line 351 “CRC crops” If CRC refers to Climate Resilient Crops, then the second “crops” is not needed done
Line 709 “The food systems approached applied in this paper” (approach rather than approached?) done
Line 887. “prof. Yusuf Byaruhanga and dr. Dorothy Nakimbugwe.” Capitalize Dr and Prof done
An overall proofread is warranted. Proofreading by a native English speaker has been performed and the English grammar has been revised on many places (not tracked in the new version of the manuscript).
Reviewer 2 Report
Interesting and well written paper, but some things but some things are repeated several time, which unnecessarily prolongs the paper; I recommend re-reading and shortening the repeated parts.
In Figure 3 it is not clear to which category belongs "not traditional" in upper part of the figure; it should be moved a bit towards the picture of bread.
Some references are not characterized well enough (for example 18, 22-24, 47, 55) and should be supplemented.
Author Response
Thank you for this peer-review. We used all your advice to improve our manuscript. Please find in red our reply to each of your points in black.
Interesting and well written paper, but some things are repeated several time, which unnecessarily prolongs the paper; I recommend re-reading and shortening the repeated parts.
We shortened the text of section 2 considerably and removed table 1 to supplementary table to improve readability. Furthermore, proofreading by a native English speaker has been performed and the English grammar has been revised on many places (not tracked in the new version of the manuscript).
In Figure 3 it is not clear to which category belongs "not traditional" in upper part of the figure; it should be moved a bit towards the picture of bread. Thank you, a text box was missing, corrected
Some references are not characterized well enough (for example 18, 22-24, 47, 55) and should be supplemented. All references – Missing details have been added.
Reviewer 3 Report
In my opinion, after the review of the state of knowledge in the Introduction, it would be worth formulating a research problem. The research problem should result from a knowledge review indicating an insufficient state of knowledge in the area in question. Only the formulation of the research problem should be a premise to identify a gap in the current state of knowledge and to present the purpose of the research. Generally, in the case of the formulated research goal, it would be worth pointing to a cognitive (scientific) goal and a utilitarian (useful) goal at the same time.
In the Abstract, I did not find a sentence that unambiguously formulates the research goal, although one of the previously mentioned goals (cognitive and / or utilitarian). The authors generally wrote down what they did. But it is not enough. If the aim of the research is formulated, then the next part can answer the question of whether the goal was achieved. In addition, you can point to new elements (observations), if the set goal could not be achieved. It is true that the authors wrote in lines 68-69 the following sentence "It requires a thorough analysis of the entire food system, which is the purpose of this paper"; however, this way of formulating the aim of the work is unacceptable when it is one of the sentences in a very long paragraph. I had to concentrate very hard to find this key sentence in the text. Reading an article should be comfortable, not a torment when I have to struggle to search for the most interesting information. I suggest creating a paragraph that will begin with the sentence: "The purpose of the study was ...".
In general, I suggest writing an article with shorter paragraphs. I suggest breaking the longest paragraphs into shorter ones. As a result, the text will be clearer and the presented information easier to analyze. Very long paragraphs become very monotonous to read, which discourages reading the article.
In the study, the authors consider the food system. It is true that Figure 1 presents a diagram of the food system, but in my opinion it would be worth giving a definition of food system. It may be the own definition of a food system or even better examples of the definition of a food system presented in the literature. In this way, the literature review would be more complete and present the main element considered in the study. For example, the following publications can be cited: "Interactions in integrated agricultural systems: The past, present and future" and "Exploring agricultural production systems and their fundamental components with system dynamics modeling". In addition, you can write about the three pillars approach in the Introduction. Here you can quote, for example, the publication "Evaluating three-pillar sustainability modeling approaches for dairy cattle production systems". It is true that the reviewed article deals with the food system, but the reference to agricultural systems may confirm the Authors' ability to show an interdisciplinary approach to the research area under consideration. In addition, there are publications in the literature that present an analysis of the food system chain based on the flow of material, energy and information. It would be worth presenting such different approaches to the analysis in the chapter Introduction. By the way, I wonder why the authors never used the word "model" in the study text? After all, the undertaken scope of the food system analysis may inspire the Authors to at least mention the possible direction of analysis related to modeling. I suggest citing a few articles about modeling in the food system.
Affordability (line: 490) and Convenience (line: 502) should appear in italics.
It would be good to explain in Table 1 what some symbols mean, for example C4 plant. Theoretically, it can be assumed that everyone knows what C4 plant are. However, this is probably a wrong assumption.
In Figure 2, the unit on the ordinate (Mtons) axis would be worth specifying more precisely, because it is probably the unit (Mtons / year). I am asking for a possible correction.
If the Authors use acronyms such as GMP, HACCP, etc., they should provide the full name on the first use. Although these acronyms were used in the text only once, it does not change the fact that their full name is given. Of course, everyone should know what the GMP and HACCP acronyms mean, but to be sure it is better to give their full names as well.
In my opinion, the Conclusions section should end with some summary sentence, and not with a listing of subsequent interventions.
Author Response
Thank you for this peer-review. We used all your advice to improve our manuscript. Please find in red our reply to each of your points in black, in the attached document
